# Liver Steatosis and Steatohepatitis Alter Bile Acid Receptors in Brain and Induce Neuroinflammation: A Contribution of Circulating Bile Acids and Blood-Brain Barrier

**DOI:** 10.3390/ijms232214254

**Published:** 2022-11-17

**Authors:** Noemi Fiaschini, Mariateresa Mancuso, Mirella Tanori, Eleonora Colantoni, Roberta Vitali, Gianfranco Diretto, Laura Lorenzo Rebenaque, Laura Stronati, Anna Negroni

**Affiliations:** 1Biomedical Technologies Laboratory, Division of Health Protection Technologies, Agenzia Nazionale per le Nuove Tecnologie, l’Energia e lo Sviluppo Economico Sostenibile (ENEA), 00123 Rome, Italy; 2Biotechnology Laboratory, Division of Biotechnologies and Agroindustry, Agenzia Nazionale per le Nuove Tecnologie, l’Energia e lo Sviluppo Economico Sostenibile (ENEA), 00123 Rome, Italy; 3Departamento Producción y Sanidad Animal, Salud Pública Veterinaria y Ciencia y Tecnología de los Alimentos, Universidad CEU-Cardenal Herrera, CEU Universities, Alfara del Patriarca, 46115 Valencia, Spain; 4Department of Molecular Medicine, Sapienza University, 00161 Rome, Italy

**Keywords:** high-fat diet, intestinal inflammation, liver, brain inflammation, bile acid receptors, bile acids, blood brain barrier

## Abstract

A tight relationship between gut-liver diseases and brain functions has recently emerged. Bile acid (BA) receptors, bacterial-derived molecules and the blood-brain barrier (BBB) play key roles in this association. This study was aimed to evaluate how non-alcoholic fatty liver disease (NAFLD) and non-alcoholic steatohepatitis (NASH) impact the BA receptors Farnesoid X receptor (FXR) and Takeda G-protein coupled receptor 5 (TGR5) expression in the brain and to correlate these effects with circulating BAs composition, BBB integrity and neuroinflammation. A mouse model of NAFLD was set up by a high-fat and sugar diet, and NASH was induced with the supplementation of dextran-sulfate-sodium (DSS) in drinking water. FXR, TGR5 and ionized calcium-binding adaptor molecule 1 (Iba-1) expression in the brain was detected by immunohistochemistry, while Zonula occludens (ZO)-1, Occludin and Plasmalemmal Vesicle Associated Protein-1 (PV-1) were analyzed by immunofluorescence. Biochemical analyses investigated serum BA composition, lipopolysaccharide-binding protein (LBP) and S100β protein (S100β) levels. Results showed a down-regulation of FXR in NASH and an up-regulation of TGR5 and Iba-1 in the cortex and hippocampus in both treated groups as compared to the control group. The BA composition was altered in the serum of both treated groups, and LBP and S100β were significantly augmented in NASH. ZO-1 and Occludin were attenuated in the brain capillary endothelial cells of both treated groups versus the control group. We demonstrated that NAFLD and NASH provoke different grades of brain dysfunction, which are characterized by the altered expression of BA receptors, FXR and TGR5, and activation of microglia. These effects are somewhat promoted by a modification of circulating BAs composition and by an increase in LBP that concur to damage BBB, thus favoring neuroinflammation.

## 1. Introduction

A crosstalk between the gastrointestinal and liver districts and central nervous system (CNS), named the liver-gut-brain axis, has recently been established, in which intestinal microbiota, enterohepatic circulation and intestinal and blood-brain barrier (BBB) integrity play key roles [1,2]. 

The gut, liver, and brain communicate principally via nerve vague and systemic circulation [3]. Bacterial metabolites, such as bile acids (BAs) or bacterial components, as the endotoxin lipopolysaccharide (LPS), can cross the intestinal barrier, reaching the circulatory stream and representing a communication channel among the intestine, liver and brain [4,5]. BAs, synthesized from cholesterol in the liver, are well known for their roles as emulsifying agents, facilitating digestion and the absorption of dietary fats and fat-soluble vitamins, being involved in energy metabolism and metabolic homeostasis [6,7,8]. BAs have also potent anti-microbial actions and thereby shape intestinal bacterial profiles. In turn, bacteria with bile salt hydrolase activity deconjugate and transform primary BAs into secondary BAs. These amphipathic molecules are also important bioactive signaling molecules binding to several nuclear and membrane receptors such as Farnesoid X receptors (FXR) and Takeda G-protein coupled receptor 5 (TGR5), both of which regulate metabolic pathways involved in lipid and glucose metabolism. These receptors, expressed mainly in the liver and intestine, have been recently associated with inflammatory processes, playing a protective role, and they are considered attractive therapeutic targets in a variety of diseases [9,10,11,12,13]. Recently, the expression of BA receptors has also been detected in mice and human brains, albeit their functions are largely unknown [14,15,16,17,18]. FXR alterations seem to disturb multiple neurotransmitter systems and alter neurobehavior, leading to the development of depression-like syndromes in mice [15,19,20]. TGR5 may have several distinct functions within the CNS and peripheral nervous system; it is expressed in both neurons and astrocytes, reduces neuroinflammation and microglial activation [21] and its activity is induced by neurosteroids and suppressed by ammonia [18].

Non-alcoholic fatty liver disease (NAFLD) and non-alcoholic steatohepatitis (NASH) represent a spectrum of hepatic diseases strictly linked to metabolic and cardiovascular disorders, such as obesity, insulin resistance, hypertension, dyslipidemia and type 2 diabetes. They are frequently recognized as the hepatic manifestation of the metabolic syndrome (MetS) [22]. These liver diseases, as well as inflammatory bowel disease (IBD), are characterized by alterations in the intestinal microbiota composition, known as dysbiosis, and by an increased intestinal permeability, favoring the translocation of BAs and other bacterial products into the bloodstream. This event can trigger systemic inflammatory phenomena even in distant locations such as the brain [23]. There is still a scarcity of studies evaluating the impact of intestinal and liver diseases on BA receptors and inflammatory response in the brain. 

Our previous study reported that mice, subjected to a high-fat and high-sugar diet (HFHSD) for 13 weeks, presented a liver steatosis, which is characteristic of NAFLD. This diet, in combination with cyclic oral dextran-sulfate-sodium (DSS) treatment that provokes an intestinal inflammation, leads to a more severe disease as NASH, which is characterized by an inflammatory infiltrate and liver fibrosis [24]. There are no currently approved pharmacological treatments for NAFLD/NASH.

The principal aim of the present study was to analyze the effect of a diet-induced NAFLD and NASH on the expression of BA receptors FXR and TGR5 in the brain and on neuroinflammation. Furthermore, alterations in BA composition and the presence of bacteria-derived harmful molecules in serum with modifications of BBB permeability were investigated as possible factors underlying neuroinflammation.

## 2. Results

### 2.1. Experimental Design

The experimental design, schematically illustrated in Figure 1, involved C57Bl/6 six-week-old male mice (male mice are more prone to develop a liver disease with respect to females that benefit from estrogen protection), divided into 3 groups: (1) standard pellet diet and tap water (CTRL group) (*n* = 7); (2) high-fat plus high-sugar diet (HFHSD group as a model of NAFLD) (*n* = 7); (3) HFHSD with cyclic administration of 1% DSS, in drinking water (HFHSD-DSS group as a model of NASH) (*n* = 7). Mice were fed with standard diet or HFHSD for the duration of the whole experiment (13 weeks), and the HFHSD-DSS group was exposed to 5 cycles of DSS. Each cycle consisted of 7-days DSS administration followed by a 10-day interval with normal drinking water. Our group has previously demonstrated, in a dose–response curve (DSS 0.5–4.0%) that oral DSS 1% for seven days determines a mild intestinal inflammation [25]. This dose, if combined with a high-fat diet, induces a moderate chronic colitis [26], so this dose was chosen for our treatments.

Induction of the acute liver injury was confirmed by the increase in biochemical markers of liver injury (plasma level of alanine and aspartate-aminotransferase) in serum (Figure A1A,B) as well as after the hematoxylin and eosin staining of liver sections.

Compared to CTRL mice, HFHSD intake increased body weight by 38%, while DSS treatment reverted this effect (Figure A1C). Morphologic analysis of HFHSD and HFHSD-DSS-fed mice revealed obvious hepatic steatosis in both groups as well as initial fibrosis and lymphocyte and neutrophil infiltration only in HFHSD-DSS mice, indicating the presence of a steatohepatitis, as described elsewhere [24].

### 2.2. Brain Expression of TGR5 and FXR Receptors Is Altered in HFHSD and HFHSD-DSS Mice

To investigate if NAFLD and NASH could alter BA receptors level even in distant organs as the brain, FXR and TGR5 expression was evaluated by IHC in various districts of the hippocampus and cortex from HFHSD and HFHSD-DSS mice and compared with standard diet fed mice. A quantitative analysis showed that the TGR5 protein was mainly localized in the cytoplasm, and its expression resulted in significantly increased (HFHSD: *p* < 0.05; HFHSD-DSS: *p* < 0.01) in some districts of Cornu ammonis (CA), such as CA1 and CA3, and the dentate gyrus (DG) of hippocampus as compared to controls (Figure 2A,B). In particular, in the HFHSD group, TGR5 expression resulted higher in CA3 as compared to the other districts, while in the HFHSD-DSS group, the highest expression was detected in the CA1 district (Figure 2C). 

On the contrary, a significant decrease in nuclear FXR level was observed, at a comparable level, in the CA1, CA3 and DG areas of the hippocampus (*p* < 0.01) only in the HFHSD-DSS group as compared to the HFHSD and control mice groups (Figure 3A). Graphics reported the average values of nuclei/mm^2^ for all the hippocampal analyzed areas in each mouse (Figure 3C).

Between the analyzed areas of the cortex, the frontal one revealed a significant overexpression of TGR5 in HFHSD (*p* < 0.05) and in HFHSD-DSS group (*p* < 0.01) as compared to control (Figure 2D), while FXR expression was significantly reduced only in HFHSD-DSS mice (*p* < 0.05) as compared to the HFHSD and control groups (Figure 3B). Graphics reported the average values of IRS for TGR5 and nuclei/mm^2^ for FXR in the frontal cortex for each mouse (Figure 2E and Figure 3C). 

### 2.3. Liver and Intestinal Injuries Provoke Alteration of BA Composition in Serum of HFHSD and HFHSD-DSS Mice

Metabolic diseases and intestinal inflammation are responsible for an impaired intestinal barrier permeability and dysbiosis. As BAs are metabolized by intestinal bacteria, their composition in the blood reflects the alterations of the intestinal microbiota. Therefore, an analysis of BA composition by LC-ESI-MS/MS was performed. The principal BAs modifications in HFHSD and HFHSD-DSS mice compared to the controls are summarized in Figure 4. According to the current literature, a prevalence of the taurine-conjugated forms was found (Figure 4A). In particular, a significant increase in those forms predominantly present in rodent was detected: tauro-beta-muricholic acid (TβMCA), known to be a potent FXR antagonist, was significantly augmented in the HFHSD-DSS mice group as compared to the HFHSD and control groups (*p* < 0.01); in addition, tauro-ursodeoxycholic acid (TUDCA), an agonist of TGR5, was also significantly increased in both the HFHSD and HFHSD-DSS groups (*p* < 0.01), while a light, although not significant, increase in tauro-chenodeoxycholic acid (TCDCA), an index of hepatic toxicity, was detected in the HFHSD group as compared to the control group (Figure 4B). On the contrary, tauro-alfa-muricholic acid (TαMCA) was reduced in both treated mice groups (*p* < 0.05), but this was in parallel with a significant decrease in its unconjugated form αMCA in the HFHSD group until its total absence in HFHSD-DSS mice. Consequently, the ratio between the unconjugated and conjugated forms of αMCA (αMCA/TαMCA), βMCA (βMCA/TβMCA) and ursodeoxycholic acid (UDCA) (UDCA/TUDCA) was significantly reduced in treated groups with respect to the control (Figure 4C). Furthermore, the secondary BAs hyodeoxycholic acid (HDCA) and deoxycholic acid (DCA) were both significantly augmented in HFHSD (*p* < 0,01) and DCA also in HFHSD-DSS mice (*p* < 0.01). Conversely, the secondary form UDCA was significantly (*p* < 0.05) increased in HFHSD vs. control and was strongly reduced (*p* < 0.01) in HFHSD-DSS mice as compared to HFHSD and control groups (Figure 4D). 

### 2.4. The Endotoxicity Marker LBP Is Increased in Serum of HFHSD-DSS Mice 

In our previous work [24], we evidenced the presence of bacterial molecules in the liver of HFHSD-DSS mice because of a leaky intestinal barrier. This finding could suggest the possibility of a systemic circulation of bacteria-derived products, which were able to reach and damage distant organs such as the brain. To verify this hypothesis, levels of LBP, a protein binding lipopolysaccharide, a major component of Gram bacteria, and a reliable serum marker of endotoxicity, was determined. Overall, data showed a strongly higher level (*p* < 0.01) of LBP in the serum of HFHSD-DSS mice and a lesser albeit not significant increase in HFHSD mice as compared to the control (Figure 5A).

### 2.5. LBP and Bile Acid Alterations Cause an Increase in BBB Permeability in HFHSD and HFHSD-DSS Mice 

Since high amounts of LBP and the augmented presence in serum of some BAs as DCA can favor an increase in BBB permeability, causing the uptake of other potentially harmful molecules in the brain, we performed an immunofluorescence analysis of the tight junction proteins ZO-1 and Occludin, whose expression negatively correlates with endothelial permeability, and of Plasmalemmal Vesicle Associated Protein-1 (PV-1), which is a reliable marker of vascular barrier injury [27], in brain sections of HFHSD and HFHSD-DSS mice. Fluorescence quantification showed a decreased immunoreactivity for ZO-1 and Occludin in brain capillary endothelial cells in the hippocampus and cortex of HFHSD-DSS (*p* < 0.01) and HFHSD (*p* < 0.05) mice as compared to control Figure 6A,B and Figure 7A,B). On the contrary, immunoreactivity for PV-1, almost absent in the CTRL group, increased in the hippocampus of the HFHSD-DSS (*p* < 0.05) group and in the cortex of the HFHSD (*p* < 0.05) and HFHSD-DSS (*p* < 0.01) groups as compared to the control group (Figure 7C,D). 

As a further demonstration of an altered BBB, we analyzed serum levels of S100β protein, a calcium-sensor protein, as a marker of altered brain permeability and glial cells activation [28]. This protein was significantly increased in the serum of HFHSD-DSS with respect to the CTRL and HFHSD groups (*p* < 0.05) (Figure 5B).

### 2.6. Neuroinflammation Marker Iba-1 Expression Is Increased in HFHSD and in HFHSD-DSS Mice 

BA receptor alterations have been recently linked to inflammatory reactions. Furthermore, a modification of BA composition and the increased serum levels of LBP and S100β, accompanied with a damaged BBB, are overall suggestive of a neuroinflammation. To investigate the inflammatory response, sections of HFHSD and HFHSD-DSS mouse brains were immunostained for Iba1, which is a marker of activated microglia (Figure 8A). Quantification of the number of Iba1+ cells showed a significant increase in Iba1 expression both in the hippocampus and in the frontal cortex of HFHSD (*p* < 0.01), and, to a greater extent, in the hippocampus of HFHSD-DSS mice (*p* < 0.001) with respect to the control group (Figure 8B).

## 3. Discussion

In our previous paper, we demonstrated that intestinal inflammation worsens liver injury induced by a high-fat and sugar diet, leading to the more severe condition of NASH and that the expression of BA receptors FXR and TGR5 is altered in inflamed intestine and liver [24]. In the present study, we demonstrated that NAFLD and, furtherly, the more severe NASH, lead to brain dysfunction characterized by a modulation of BA receptors FXR and TGR5. These alterations are supposed to be induced by an imbalance of BA composition in serum, and they are favored by a systemic exposure to the bacterial endotoxin LPS. Both these conditions provoke a BBB destruction, which is assessed by ZO-1, Occludin, PV-1 and S100β alterations, leading to neuroinflammation, as evaluated by measuring Iba-1 expression.

More in detail, we found a significant down-regulation of nuclear FXR in the hippocampus and cortex of HFHSD-DSS mice as compared to the HFHSD and control groups, which is a variation that is consistent with inflammatory conditions. Indeed, FXR is known to be negatively regulated by inflammatory molecules as cytokines and pro-inflammatory transcription factors as NF-κB [29]. On the contrary, we observed an up-regulation of cytoplasmic TGR5 expression, according with results reporting TGR5 overexpressed in the immune response, in activated macrophages and microglia (the brain resident immune cells) during the initial phase of neuroinflammation, probably as an attempt to protect the brain [21,30]. This agrees with the alteration of TGR5 detected in HFHSD mice other than in HFHSD-DSS mice, where brain inflammation is in an initial phase, and with the attenuation of FXR signals only in HFHSD-DSS mice, where brain inflammation is overt. The expression of the two receptors, extensively described in the liver and gut, has only recently reported in the brain as a regulator of metabolic/energy homeostasis or related to neurological decline [20,21,31,32,33,34,35]. However, to our knowledge, very few studies have correlated their modifications in the brain with metabolic dysfunctions linked to diet-induced liver diseases: Jena and colleagues, for instance, reported that a fructose, palmitate, and cholesterol (FPC)-enriched diet reduced FXR and TGR5 signaling in the brain after eight months of treatment and induced a neuroinflammation [36,37], whereas Czarnecka et al. [35] showed a buildup of BAs in the blood and a reduction in the expression of mRNAs coding for FXR in the hippocampus and cerebellum of rat with thioacetamide-induced acute liver failure.

We wondered about the possible causes of these findings: the gut, liver and brain communicate through signaling via nerve vague and via systemic circulation. From the gut, bacteria have been shown to send signals to the CNS via their metabolites, among which BAs impact the behavior and brain function of the host organism [38]. It is now believed that the BAs signal from circulation reaches the brain after crossing the BBB. Since the composition of the serum BAs pool may be regulated by the action of intestinal bacteria, it is possible that BAs represent a direct link between the gut microbiome and the brain [4]. It is well known that approximately 95% of the BAs, released into the intestine as primary acids and transformed by the gut bacteria into secondary acids, are reabsorbed into the portal system via the enterohepatic circulation and are recycled into hepatocytes. In disease states involving intestinal and liver injury, a disruption of this reuptake, together with changes in the BAs composition due to dysbiosis, induce a spillover of BAs into the circulation that can produce a variety of pathological effects, also in distant organs such as the brain. Indeed, in the last decades, important new insights have been gained, proposing BAs signaling as important determinants in a variety of neuropathological conditions, such as Alzheimer’s disease [39] Parkinson’s disease [13], and hepatic encephalopathy [11,40]. 

According to the literature, we found that the taurine-conjugated forms of BAs were prevalent in the serum and in particular, some components such as TβMCA and TUDCA were significantly increased in treated mice as compared to controls, as a manifestation of hepatocytes damage, thus suggesting a reduced deconjugating bacterial activity in the gut. 

Interestingly, variations of these two Bas are compatible with the brain receptor modifications: indeed, TβMCA and TUDCA are, respectively, an antagonist of FXR and agonist of TGR5. Thus, their increased presence in the serum could in part explain the alterations of the two receptors in the brain. In this sense, the increase in TβMCA only in the HFHSD-DSS mice group overlapped with a diminished presence of FXR in the same group, and the significant increase in TUDCA in both treated groups is consistent with the elevated levels of TGR5.

Furthermore, recent literature reports that high levels of circulating serum BAs as DCA, HDCA and CDCA interfere and disturb gap junction function, leading to an increased permeability of the BBB and consequently allowing for BAs and other molecules to diffuse into the brain [5,41]. Meanwhile, UDCA, with a recognized anti-inflammatory, antitumoral and intestinal membrane permeability protection activity [42], exerts a protective effect on neurons and brain endothelial cells. Intriguingly, either DCA and HDCA were increased in the mice of this study, while UDCA was reduced only in HFHSD-DSS as compared to the HFHSD and control groups, providing a causal relationship between BAs alterations and BBB damage. The latter finding is also indicative of a different microbiota composition between the two treated groups. It is worth noting that in addition to the BBB damage, HDCA and DCA are reported to be associated with obesity and colon rectum cancer in mice, being both increased in HFHSD and HFHSD-DSS mice [43,44].

In our previous work, we showed, in mice with diet-induced liver injury, an impaired intestinal barrier that could permit the translocation of harmful molecules to blood circulation. In keeping with this finding, our present data revealed also a significant strong increase in the LBP concentration in the serum of HFHSD-DSS mice and, to a lesser extent, in HFHSD mice. In this perspective, a systemic exposure to LPS, as a result of a damaged intestinal barrier, might be responsible for a BBB dysfunction. 

In agreement with these considerations, our experiments showed a decrease in the tight junction proteins ZO-1 and Occludin and an increase in the marker of vascular barrier damage, PV-1, both in the hippocampus and cortex of HFHSD and, to a greater extent, in HFHSD-DSS mice as compared to the control group, suggesting the presence of a BBB injury that could facilitate the uptake of toxic molecules. This finding was also supported by the circulating increased level of S100β, which is a calcium-sensor protein that impacts multiple signal transduction pathways [45]. It is a glial cells activation marker widely considered to be an important biomarker for several neuronal diseases as well as BBB breakdown [28]. 

All these evidence strongly suggested that an aberrant BAs composition linked to a strong increase in circulating LPS can concur to damage the BBB, favoring a neuroinflammation. To assess neuroinflammation, an analysis of Iba-1, a marker of activated microglia, the primary immune cells in the CNS, was carried out. It is known that when a danger signal is detected, microglia undergo a rapid change in morphology and function, which is a process that has been termed activation, consisting of the retraction and thickening of the processes, a size increase in the cell body and the excretion of cytokines by the cells. As expected, immunostaining showed an increase in the expression of pro-inflammatory marker Iba-1 in the hippocampus and cortex of the HFHSD-DSS group, while a minor but significant increased expression was detected also in HFHSD mice as compared to the control group, confirming the condition of inflammation, already hypothesized as a result of BA receptor analysis. Overall, our data agree with recent papers reporting the effects of a high-fat diet in increasing BBB permeability, oxidative stress and neuroinflammation [40,46,47,48]. 

In summary, the present study recognizes a relationship between liver injuries induced by a dysregulated diet, such as NAFLD and NASH, which are both closely associated to MetS, and brain dysfunctions characterized by an alteration of BA receptors FXR and TGR5 and an activation of the microglia. These effects are somewhat promoted by a modification of circulating BAs composition and by an increase in LBP and S100β in blood that concur to increase BBB permeability, thus favoring neuroinflammation. We plan to investigate the contribution of these receptors to brain functions more deeply in the future also by using specific BAs as agonists or antagonists. This study constitutes a first step toward a better comprehension of the relationship between gut–liver diseases, circulating BAs and their receptors in brain, the so-called gut–liver–brain axis, and it is now becoming clear that dietary and therapeutic interventions targeting this axis represent future promise for many gastrointestinal and neurological disorders.

## 4. Materials and Methods

### 4.1. Mouse Diet

C57BL/6J six-week-old male mice from Envigo RMS, Srl.(Indianapolis, IN, United States) were housed in collective cages at 22°C ± 1 °C under a 12 h light/dark cycle and with food and water provided ad libitum. After a 1-week acclimatization, animals were randomly divided into 3 groups as reported in the experimental scheme in Figure 1. High-fat diet (18% kcal protein, 24% kcal carbohydrate, 58% kcal fat for a total of 5.6 kcal/g) was provided by Laboratory Dottori Piccioni, Milan, Italy). High sugar (23.1 g L-fructose plus 18.9 g L-glucose/L) (Sigma-Aldrich, St. Louis, MO, USA) was added to tap water; the cyclic administration of 1% (*w*/*v*) DSS (molecular mass, 36,000–50,000 Da, MP Biomedicals, Santa Ana, CA, USA) was also added in drinking water. After 13 weeks, animals were weighed and euthanized, brains were removed for immunohistochemistry and immunofluorescence, and serum samples were stored at −80 °C for biochemical analysis.

### 4.2. Ethical Statement

All animal experiments adhered to the ARRIVE (Animal Research: Reporting on In Vivo Experiments) guidelines and were approved by the Ministry of Health for the protection of animals used for experimental purposes (n.266/2019-PR prot. EE25E.10), and the study was conducted in accordance with Italian regulations on animal welfare. The protocol was approved by the Committee on the Ethics of Animal Experiments of the Italian National Agency for New Technology, Energy and Sustainable Economic Development (ENEA). 

### 4.3. Immunohistochemistry (IHC)

Brains were fixed in 10% formalin and embedded in paraffin for routine histology. Here, 4 μm sections were mounted on slides and stained with standard hematoxylin and eosin (H&E) techniques. For IHC, sections (4 μm) of paraffin-embedded brains were prepared following standard protocol. Briefly, sections were dewaxed for 20 min at 56 °C and incubated in citrate buffer pH 6.0 for 20 min at 95 °C. Afterward, sections were washed in water for 5 min, and peroxidases were then inhibited by incubating sections in 3% H_2_O_2_ for 10 min. Sections were treated with 5% bovine serum albumin (BSA) (Santa Cruz Biotechnology Inc., Santa Cruz, CA, USA) for 20 min and incubated with primary antibodies anti-FXR, (1:100, Perseus Proteomics, Tokyo, Japan), TGR5 (1:500, AbCam, Cambridge, UK), and Iba-1 (1:500, Wako Pure Chemical Industries, Osaka, Japan) antibodies, and then diluted in phosphate-buffered saline (PBS) for 1 h at room temperature in a moist chamber. Sections were washed in PBS, incubated for 30 min with the secondary anti-rabbit antibody and washed again in PBS. The DAB detection kit (Dako North American Inc., Carpinteria, CA, USA) was used to visualize the antigen. Finally, sections were counterstained with hematoxylin. 

All stained sections were captured with a Leica digital camera. Quantitative analysis was performed blind in 4 randomly selected brain sections. To evaluate the microglia activation status in the hippocampus and in cortex, immunohistochemical stain was measured counting positive cells for Iba-1 by the imaging software NIS-Elements BR 4.00.05 (Nikon Instruments S.p.A., Florence, Italy). The quantification of Iba-1 and FXR density was expressed as the number of stained cells per area (mm^2^). For TGR5 analysis, the manual scoring procedure with digital images of different regions of the hippocampus (DG, CA1, CA2 and CA3) and the cortex (frontal, medial and caudal cortex) was performed for each animal. The immunoreactivity score (IRS) was determined by two independent observers, who assessed the relative amounts of stained cells and staining intensity. An ordinal scale was established based on the number of stained cells and the staining intensity in each region of interest: 0 was defined as no stain or weak stain in <10% of the cells; 1 was defined as a weak stain in ≥10% of cells; 2 was defined as a moderate stain in ≥10% of the cells; and 3 was defined as a strong stain in ≥10% of the cells. TGR5 high expression was defined as an intensity of 2 or 3 in ≥10% of the cells. Graphics reported the media values of IRS for each group with standard deviation.

### 4.4. Immunofluorescent Staining

Paraffin-embedded brain sections of 4 μm were rehydrated, blocked with 5% BSA in PBS with 0.3% Triton X-100 and incubated with the primary FITC conjugate anti-ZO-1 and anti-Occludin (1:100, ThermoFisher Scientific, Waltham, MA, USA) and anti-PV-1 (1:100, BD Pharmingen TM, San Diego, CA, USA) for 24 h at 37 °C, and washed for three times. Thereafter, sections incubated with anti-Occludin antibody were treated with Goat anti-Rabbit IgG (H+L) Superclonal™ Secondary Antibody, Alexa Fluor^®^ 488 conjugate (1:2000, ThermoFisher Scientific) and sections incubated with anti-PV-1 antibody were treated with Cy™3 AffiniPure Donkey Anti-Rat secondary antibody (1:200) (Jackson ImmunoResearch Europe Ltd., Ely, Cambridgeshire, UK) for 2 h at 37 °C and washed for three times. Nuclei were counterstained with 4′6-diamidino-2-phenylindole (DAPI). Fiji (ImageJ) software package (NIH, National Institutes of Health) was used for image analysis and fluorescence intensity quantification. 

### 4.5. Enzyme-Linked Immunosorbent Assay (ELISA)

Lipopolysaccharide binding protein (LBP) and S100β quantification in mice serum was performed using ELISA kits (Enzo LBP ELISA kit and Abcam ELISA kit, respectively) according to the manufacturer’s instructions.

### 4.6. Bile Acid Profiling

Liquid Chromatography coupled to Electrospray Ionization Tandem Mass Spectrometry (LC-ESI-MS/MS) analysis of the BAs was performed as previously described [49,50] on BA samples extracted from mice serum. Briefly, proteins in 25 μL of plasma were precipitated by adding 100 μL of a methanol:acetonitrile mixture (5:3 *v*/*v*), and they were incubated on ice for 15 min. Samples were centrifuged at 10,000× *g* for 40′ at 15 °C, and 5 μL of supernatant was injected into an LC-ESI-MS/MS system. Formononetin (5 μg/mL) was added as the internal standard to all samples. Each experimental group consisted of 4 animals each. Liquid chromatography (LC) was carried out using a Phenomenex C18 Luna column (100 × 2.0 mm, 2.5 μm), and the mobile phase was composed by water–0.1% formic acid (A) and acetonitrile–0.1% formic acid (B). The gradient was: 95%A:5%B (1 min), a linear gradient to 25%A:75%B over 40 min, 2 min isocratic, before going back to the initial LC conditions in 18 min. Five μL of each sample was injected, and a flow of 0.25 and 0.8 mL was used throughout the LC semi-polar runs. MS analysis was performed using a quadrupole-Orbitrap Q-exactive system (ThermoFisher scientific, USA), operating in positive/negative heated electrospray ionization (HESI) coupled to an Ultimate HPLC-DAD system (Thermo Fisher Scientific, Waltham, MA). Mass spectrometer parameters were as follows: capillary and vaporizer temperatures 30 °C and 270 °C, respectively, discharge current of 4.0 KV, probe heater temperature at 370 °C, and S-lens RF level at 50 V. The acquisition was carried out in the 110/1600 *m/z* scan range with the following parameters: resolution 70,000, microscan 1, AGC target 1 × 10^6^, and maximum injection time 50. Full scan MS with data-dependent MS/MS fragmentation was used for metabolite identification. Metabolites were quantified in a relative way by normalization on the internal standard amount. The targeted identification of BAs was performed by comparing chromatographic and spectral properties with authentic standards.

### 4.7. Statistics

Experiments were repeated 3 times. Data are expressed as the mean ± SD. Statistical analyses were performed with GraphPad Prism Software (GraphPad Software, San Diego, CA, USA). Differences were analyzed by one-way analysis of variance (ANOVA) and unpaired Student *t*-test. *p* values < 0.05 were considered to indicate significance. 

## Figures and Tables

**Figure 1 ijms-23-14254-f001:**
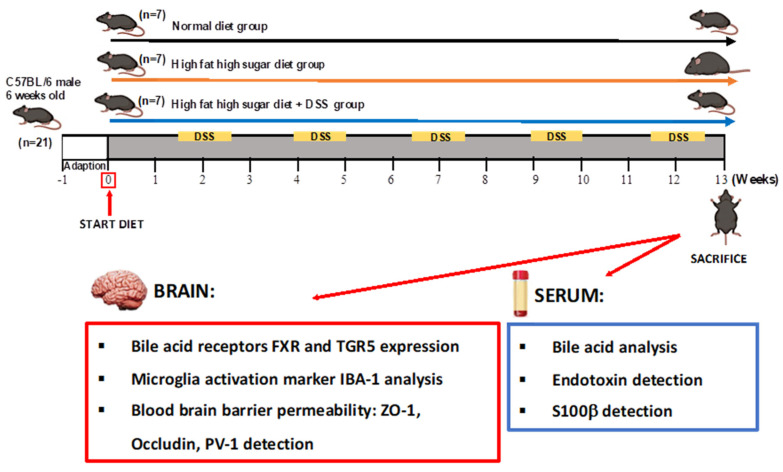
Experimental design scheme. Animals were randomly divided into 3 groups: (1) standard pellet diet and tap water (CTRL group) (*n* = 7); (2) high fat diet plus high sugar (HFHSD group as a model of NAFLD) (*n* = 7); (3) HFHSD with cyclic administration of 1% (*w*/*v*) DSS, in drinking water (HFHSD-DSS group as a model of NASH) (*n* = 7). Mice were fed with standard diet or HFHSD for the duration of the whole experiment, and the HFHSD-DSS group was exposed to 5 cycles of DSS. After 13 weeks of treatment, mice were sacrificed and brains were collected for BA receptors, Iba-1, ZO-1, Occludin and PV-1 analyses and serum was collected for BAs, S100β protein and endotoxin analyses.

**Figure 2 ijms-23-14254-f002:**
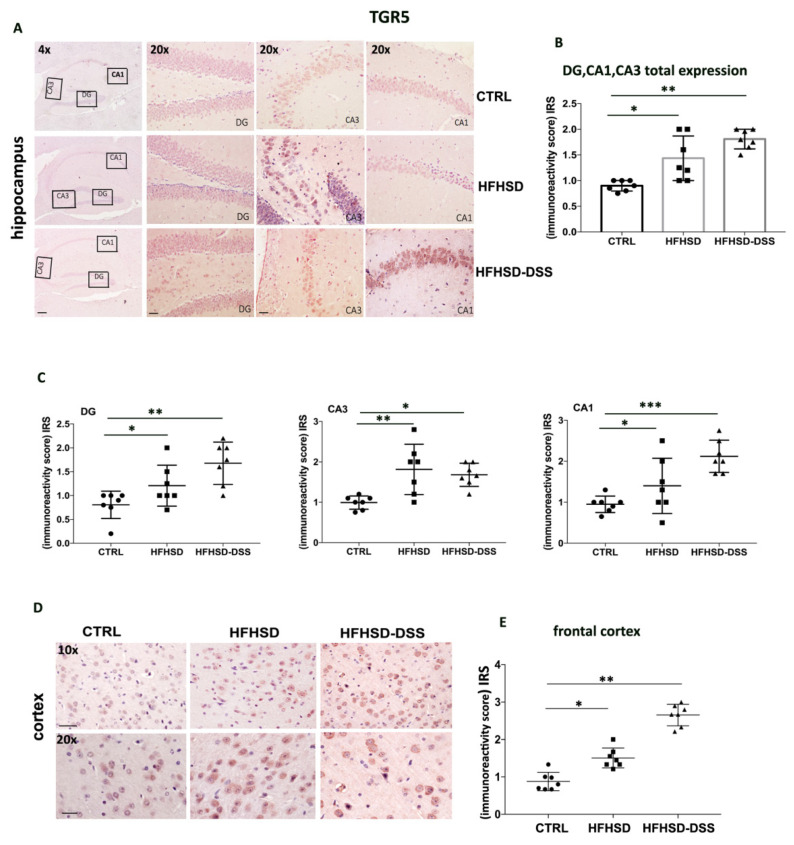
Representative images of immunohistochemistry (IHC) staining of TGR5 in different regions of the hippocampus (**A**) and in the frontal cortex (**D**) displaying the expression and cytoplasmic localization of TGR5. 4× magnification: Scale bar = 100 μm. 10× magnification: scale bar = 50 μm; 20× magnification: Scale bar = 25 μm. Graphic (**B**) reported the average values of immunoreactivity score (IRS) for all analyzed areas. Graphics (**C**) show TGR5 quantification for each district. Graphic (**E**) represent IRS in frontal cortex. Mean ± SD for each group is shown. Statistical significance: * *p* < 0.05; ** *p* < 0.01; *** *p* <0.001.

**Figure 3 ijms-23-14254-f003:**
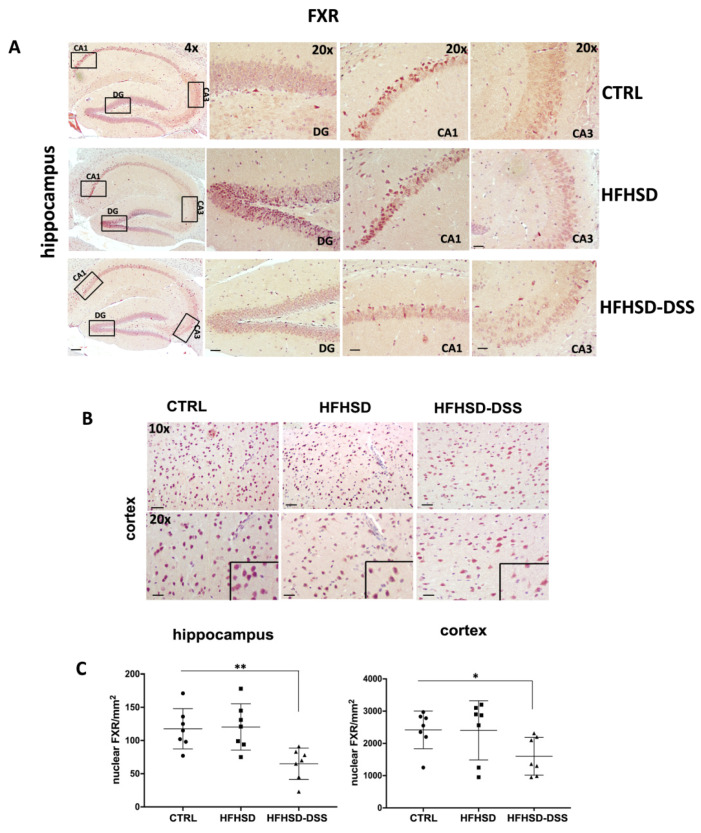
Representative images of immunohistochemistry (IHC) staining of FXR in different regions of hippocampus (**A**) and in frontal cortex (**B**) displaying expression and nuclear localization of FXR. 20× magnification: scale bar = 25 μm. 10× magnification: scale bar = 50 μm. 4× magnification: scale bar 100 μm; Quantitative analysis of FXR expressing nuclei /mm^2^ in both brain districts (**C**). Mean ± SD for each group is shown. Statistical significance: * *p* < 0.05; ** *p* < 0.01.

**Figure 4 ijms-23-14254-f004:**
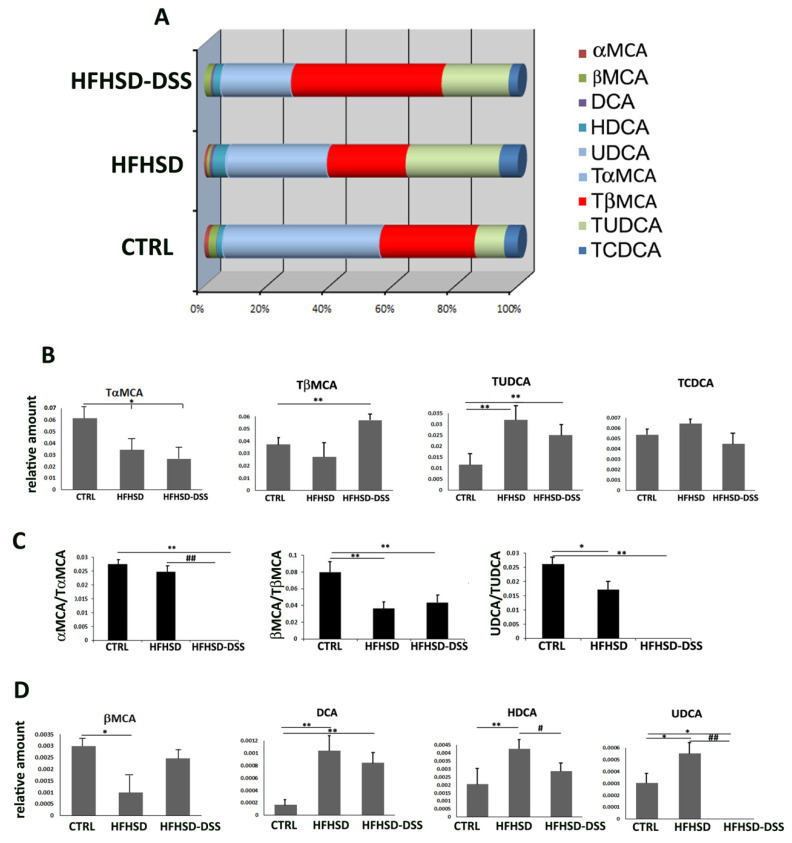
(**A**) Composition of principal BAs in serum of CTRL, HFHSD and HFHSD-DSS mice after 13 weeks of treatment; (**B**) Relative quantitative analysis of the most altered conjugated BAs in the three groups of mice; (**C**) Ratio between free/conjugated form of αMCA, βMCA and UDCA bile acids; (**D**) Relative quantitative analysis of the most altered unconjugated bile acids in the three groups of mice. Each value is presented as the mean ± SD for four mice. * Treated groups vs. CTRL; # = HFHSD vs. HFHSD-DSS; Statistical significance: * *p* < 0.05; ** *p* < 0.01; # *p* < 0.05; ## *p* < 0.01.

**Figure 5 ijms-23-14254-f005:**
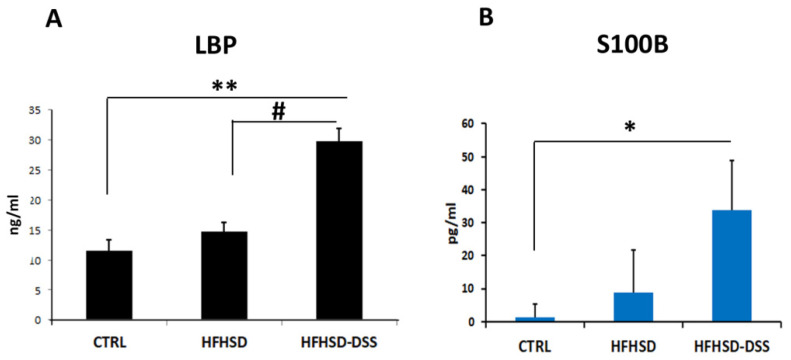
(**A**) Lipopolysaccharide binding protein (LBP) and (**B**) S100β protein concentrations in serum of CTRL, HFHSD and in HFHSD-DSS mice, after 13 weeks of treatment; Statistical significance: * *p* < 0.05; ** *p* < 0.01; # *p* < 0.05.

**Figure 6 ijms-23-14254-f006:**
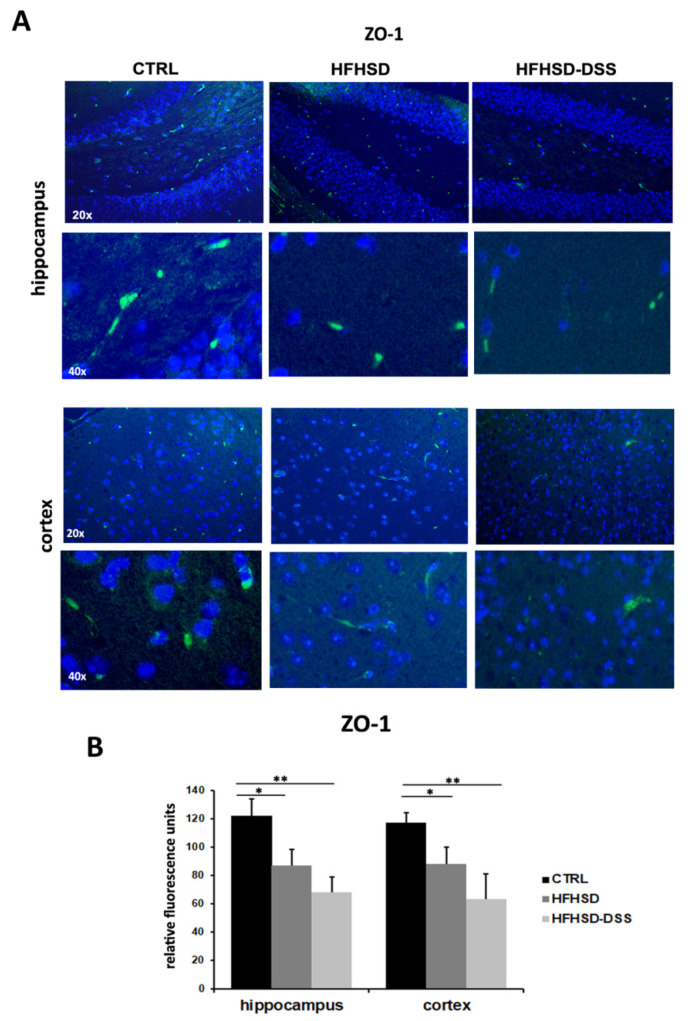
(**A**) Representative images of ZO-1 immunofluorescence in the hippocampus and cortex of both treated groups and control. (**B**) Quantitative analysis of immunofluorescence detection of ZO-1 in the hippocampus and cortex of both treated groups compared to controls; each value is presented as the mean ± SD for seven mice. Statistical significance: * *p* < 0.05; ** *p* < 0.01.

**Figure 7 ijms-23-14254-f007:**
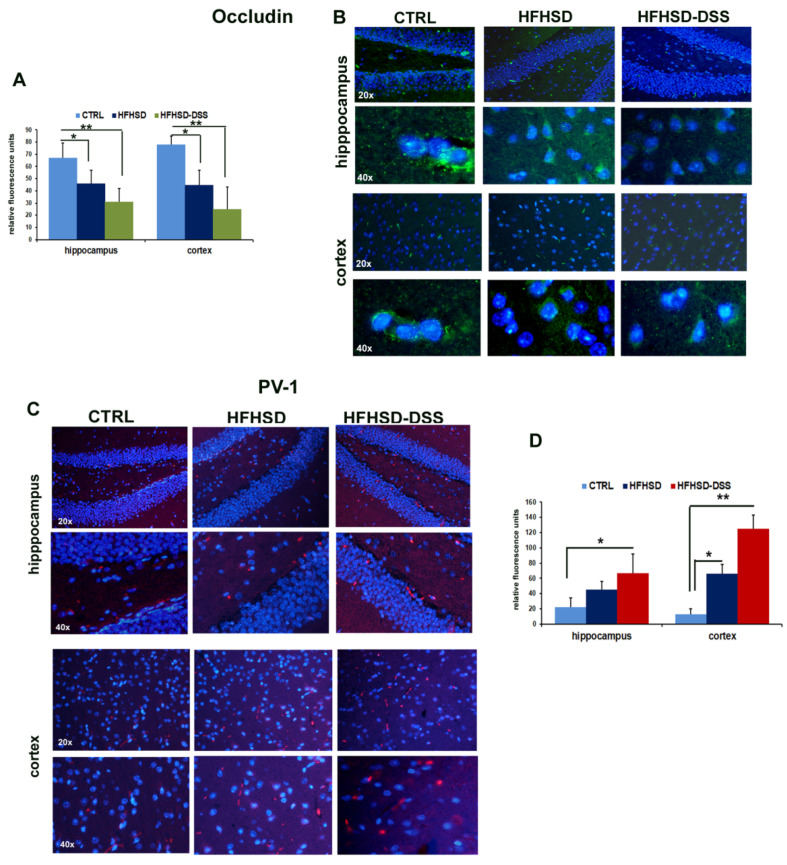
Representative images of Occludin (**B**) and PV-1 (**C**) immunofluorescence in hippocampus and cortex of both treated groups and control. Quantitative analysis of immunofluorescence detection of Occludin (**A**) and PV-1 (**D**) and in the hippocampus and cortex of both treated groups compared to controls; each value is presented as the mean ± SD for seven mice. Statistical significance: * *p* < 0.05; ** *p* < 0.01.

**Figure 8 ijms-23-14254-f008:**
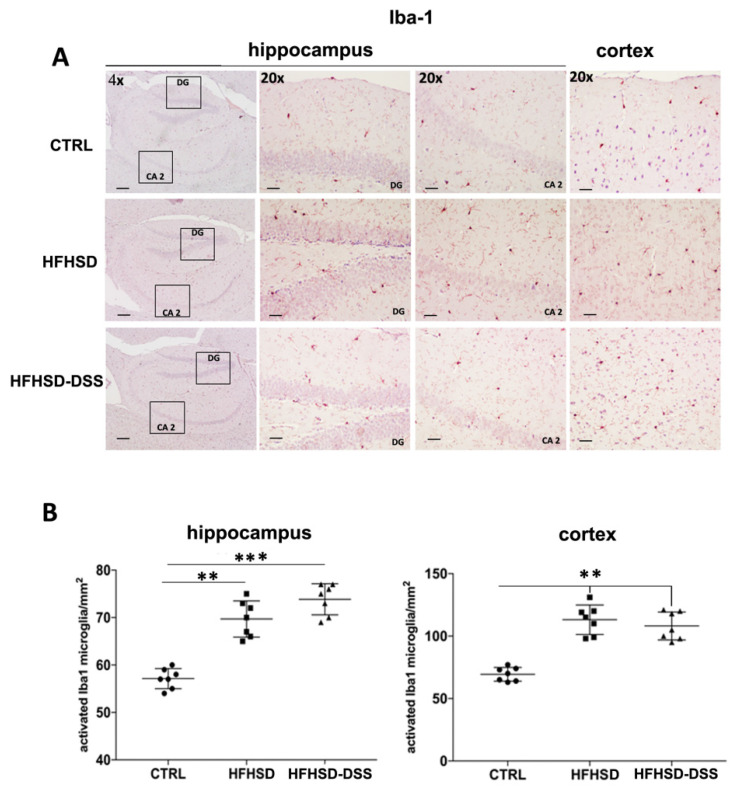
(**A**) Representative images of immunohistochemistry (IHC) staining of Iba-1 in different regions of the hippocampus and frontal cortex of treated mice as compared to control. 20× magnification: Scale bar = 25 μm. 4× magnification: scale bar 100 μm; (**B**) Quantitative analysis of stained nuclei/mm^2^. Each value is presented as the mean ± SD for seven mice. Statistical significance: ** *p* < 0.01; *** *p* < 0.001.

## Data Availability

Data analyzed during the study are available from the corresponding authors on reasonable request.

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
