# Peer review of "Liver Steatosis and Steatohepatitis Alter Bile Acid Receptors in Brain and Induce Neuroinflammation: A Contribution of Circulating Bile Acids and Blood-Brain Barrier"

_ijms, 2022, doi:10.3390/ijms232214254_

Round 1

Reviewer 1 Report

1. How did the authors determine the DSS dose (1%) ???

2. Show the dose responses of DSS (from 0.1 - 2%), if 1% DSS showed the beneficial effects. 

3. Confirm the ELISA data by using RT-PCR or western blotting. 

4. Why the authors used only male mice?

Author Response

  1. How did the authors determine the DSS dose (1%) ???

Response: Cyclic oral dose of DSS 1%, in combination with a high fat diet, has been already used by some Authors to determine a chronic colitis that increases inflammation and fibrogenesis in non-alcoholic fatty liver disease mouse model to accelerate the progression towards the more severe steatohepatitis (Gäbele E, et al.  2011 J Hepatol. 2011;55:1391-9.; Cheng C. et al. Life Sci. 2018;209:157-166; Shu X, et al. Evid Based Complement Alternat Med. 2017;2017:9208314).

  1. Show the dose responses of DSS (from 0.1 - 2%), if 1% DSS showed the beneficial effects.

Response: Our group has previously performed a dose response of DSS (0.5-4.0%) to determine gut damage and demonstrated that oral DSS 1% for seven days determines a mild intestinal inflammation (Palone et al. IBD 2014). This dose, if combined with a high fat diet, has been previously demonstrated to induce a moderate colitis (Li X et al. Am J Physiol Gastrointest Liver Physiol. 2019; Tanaka S, et al. Biochem Biophys Res Commun. 2020), so it was chosen for a longer treatment, as in this work.

  1. Confirm the ELISA data by using RT-PCR or western blotting..

Response: We apologize with the reviewer, but we cannot perform a RT-PCR neither a western blot, due to the lack of a sufficient serum quantity.  We have already used all serum to perform LC-ESI-MS/MS analysis and the two ELISAs. However, ELISA is considered the best quantitative method to detect a protein and we want to assure this reviewer that the assays used in this paper, commonly used by our and other groups, are validated and demonstrated to give reliable results.

  1. Why the authors used only male mice?

Response: We used male mice as they are more prone to develop a liver disease as respect to female that are relatively resistant due to the protective action of estrogens. (Montefusco D, et al. Sphingosine kinase 1 mediates sexual dimorphism in fibrosis in a mouse model of NASH. Mol Metab. 2022 Aug;62:101523. )

Reviewer 2 Report

The authors characterized the alterations in the brain expression of bile acid (BA) receptors in animal models of liver steatosis and steatohepatitis and estimated only a few immunofluorescence markers of neuroinflammation and blood-brain barrier (BBB) dysfunction in these disorders. Overall, this study is descriptive and too simplistic. The manuscript must have a greater scientific value to provide a valuable contribution to the field. The authors did not show the contribution of BBB to neuroinflammation or alterations in serum BA content to the BA receptor expression in the brain but simply explored these parameters and tried to connect them in the Discussion. To provide evidence of contribution, the authors would need to use gain- and loss-of-function experiments. Furthermore, the characterization of BBB dysfunction is very limited.  The tight junction proteins include claudin-5, occludin, and zonula occludens such as ZO-1. Decreased expression of the tight junction protein ZO-1 may not necessarily lead to the BBB breakdown. Commonly utilized methods to estimate BBB integrity in animal models include the ex vivo measurement of cerebral extravasation of exogenous dyes such as fluorescein and Evans Blue that are injected intravenously.  There should be a more thoughtful and extended experimental design with a mechanistic approach (knockout models or antagonist/agonist administration) to focus on the potential contribution of FXR and TGR5 receptors to BBB dysfunction and neuroinflammation in liver steatosis and steatohepatitis. In the current manuscript version, molecular mechanisms are suggested but not shown.  

Author Response

The authors characterized the alterations in the brain expression of bile acid (BA) receptors in animal models of liver steatosis and steatohepatitis and estimated only a few immunofluorescence markers of neuroinflammation and blood-brain barrier (BBB) dysfunction in these disorders. Overall, this study is descriptive and too simplistic. The manuscript must have a greater scientific value to provide a valuable contribution to the field. The authors did not show the contribution of BBB to neuroinflammation or alterations in serum BA content to the BA receptor expression in the brain but simply explored these parameters and tried to connect them in the Discussion. To provide evidence of contribution, the authors would need to use gain- and loss-of-function experiments. Furthermore, the characterization of BBB dysfunction is very limited.  The tight junction proteins include claudin-5, occludin, and zonula occludens such as ZO-1. Decreased expression of the tight junction protein ZO-1 may not necessarily lead to the BBB breakdown. Commonly utilized methods to estimate BBB integrity in animal models include the ex vivo measurement of cerebral extravasation of exogenous dyes such as fluorescein and Evans Blue that are injected intravenously.  There should be a more thoughtful and extended experimental design with a mechanistic approach (knockout models or antagonist/agonist administration) to focus on the potential contribution of FXR and TGR5 receptors to BBB dysfunction and neuroinflammation in liver steatosis and steatohepatitis. In the current manuscript version, molecular mechanisms are suggested but not shown.

Answer:  

We thank the Reviewer for the useful criticism and the precious suggestions to our work. However, we would point out that this work was not aimed to study the role of BA receptors FXR and TGR5 in brain, but  the main goal was to investigate if a steatohepatitis, the hepatic manifestation of metabolic syndrome, induced by a western diet combined with an intestinal inflammation, can lead, as a consequence, to an imbalance of FXR and TGR5 bile acid receptors also in brain, other than in gut and liver, and modifications of serum bile acids composition compatible with the alteration of these two receptors.  Concomitantly, detection of an increased LBP serum levels suggested a systemic inflammation that could damages BBB, that was confirmed by ZO-1 analysis in brain and S100b in serum.  As the Gadaleta RM and Biagioli M. groups demonstrated that down-regulation of FXR and up-regulation of TGR5 are indicative of an inflammatory condition in other systems, as intestinal mucosa and immune cells, microglia activation was used to confirm this inflammatory status.

We retain that this work represents a pilot study, not only descriptive, that recognizes a relationship between metabolic syndrome, induced by a deregulated diet, and brain BA receptor alterations and neuroinflammation.  It has been recently emerged that metabolic dysregulation together with dysbiosis are central in the pathogenesis of diseases of the so-called gut-liver-brain axis and it is now becoming clear that dietary and therapeutic interventions targeting this axis, represent future promise for many gastrointestinal and neurological disorders.  Very scarce studies exist on this topic and are all mentioned and commented in the discussion. We plan in the next future to deep investigate the possible role of these receptors in brain, by using specific BA receptor agonists or antagonists, chosen among those resulted modified in this work, as suggested by Reviewer.   After all, the study on role of BA receptors in brain is just at the onset and mainly associated with degenerative or cognitive diseases, albeit with controversial results and their role is not still well elucidated.

The above considerations have been added in the discussion.

As suggested by Reviewer, in order to better characterize BBB damage, we performed immunofluorescence experiments with Occludin and Plasmalemmal Vesicle Associated Protein-1 (PV-1), a marker of a damaged vascular barrier, obtaining a further confirmation of the barrier injury in particular in DSS-treated mice.

We added a new figure (Figure 7) showing the results of these experiments.

English form has been carefully checked by a native English-speaker.

Round 2

Reviewer 2 Report

The revised manuscript is fairly more elaborated than the original version. Most importantly, the authors pointed out that this is a pilot study with a focus on the liver-gut-brain axis, a relatively new field. Adding new blood-brain barrier damage markers undoubtedly improved the manuscript. The English language has improved a lot after revision as well. However, I still suggest some comments for revisions.

Strengths of the study:

The results of this study suggest that hepatic steatosis and steatohepatitis can have significant effects on the brain through the bile acid signaling pathway. In particular, the study shows thatcompromised brain function expressed as blood-brain barrier injury and microglial activation in the hippocampus and cortex may result from alterations in bile acid metabolism and its associated receptors. Dysfunction within these brain regions is often associated with cognitive deficits. Overall, the presented study highlights the importance of protecting the liver to maintain brain health and reveals new aspects of the liver-brain axis.

Minor revisions to the manuscript are needed:

1.        In the Abstract and Introduction, dextran sulphate sodium (DSS) has been replaced with sodium dextran sulfate (lines 26-27, line 81). However, this change was unnecessary since the authors use DSS and not SDS throughout the text. The only mistake that needed to be replaced was “sulphate”. The correct word is “sulfate”.

2.        There should be more information on why this study is important. In their response to the comments, the authors state that this study “recognizes a relationship between metabolic syndrome, induced by a deregulated diet, and brain BA receptor alterations and neuroinflammation.” However, this rationale is missing in the manuscript. Therefore, it would be great to mention metabolic syndrome and nonalcoholic fatty liver disease (NAFLD) in the manuscript. In the current version, the authors only mentioned NASH, an aggressive form of NAFLD, but not NAFLD in the manuscript.There are no currently approved pharmacological treatments for NASH/NAFLD. This could be also mentioned in the Introduction to further highlight the importance of this research topic.

3.        If HFHSD-DSS mice represent the model of NASH, then do HFHSD mice represent a model for NAFLD? It needs to be clarified. 

4.        P. 3, line 124: The authors use “Dentatus gyrus”. It would be more appropriate to use either of these terms: gyrus dentatus (Latin) or dentate gyrus (more common).

5.        P. 3, lines 123-124: “…in some districts of Cornu Ammonis (CA)….of hippocampus as compared to controls (Figure 2A and C)”. Could you please clarify what exact hippocampal regions of HFHSD and HFHSD-DSS mouse brains had higher TGR5 expression as compared to the control group? Please also address if Fig. 2C contains data for the sum of TGR5 expressing cells in all three analyzed hippocampal regions or is it an average cell number. Would it not be better to provide graphs for each hippocampal area in Fig. 2C?

6.        P. 4, lines 133-134: “was observed in the CA1, CA3 and DG areas of hippocampus….”. Could you please clarify if FXR levels were significantly higher in all three hippocampal regions? Fig. 3A contains images for different hippocampal areas but the quantification (Fig. 3C) has been performed without distinguishing the areas (Fig. 3C).

7.        What exactly cortex area has been analyzed for evaluating TGR5 and FXR expression levels (Figures 2 and 3)? Was it frontal, medial, or caudal? These cortical regions are mentioned in the Materials and Methods (p. 13, line 404).

8.        It would be better to use HFHSD mice instead of “steatosis affected mice” and HFHSD-DSS mice instead of “steatohepatitis affected mice” for subtitles for consistency and correctness (lines 16-117, 145-146, 181 – subtitles 2.1-2.3).

9.        Subtitle 2.5 (p. 9, line 226) needs to be changed. It is well known that Iba1 is a pan-microglial marker whose expression increases with microglial activation. Did authors mean “Alterations in neuroinflammation marker Iba-1 expression in the brains of HFHSD and HFHSD-DSS mice”?

10.     It seems like the full name for UDCA (p.6, line 165) is omitted.

11.     P.7, lines 201-202: It should be endothelial (not epithelial) permeability if the context is BBB. Besides, it would be better to include a reference confirming that PV-1 is a reliable marker of vascular brain injury.

12.     It would be good to include reference demonstrating that serum S100b levels can serve as a marker of altered BBB permeability and glial cell activation (p. 9, lines 223-224).

13.     P. 10, lines 251-252: It would be better to delete “a condition of” since neuroinflammation is a condition itself (characterized by the elaboration of proinflammatory mediators). Besides, “evaluated by the activation of Iba-1” should be replaced with “evaluated by measuring Iba-1 expression”. Iba-1 is increased in activated microglia (it is not activated).

14.     It would be more precise to use “To assess neuroinflammation” instead of “To assess this hypothesis” (p.12, line 336).

15.     P.12, line 350: It is more correct to use “an increase in the expression of proinflammatory marker Iba-1”. Microglia can be activated but not the marker.

16.     Some sentences are too long and might need to be divided into shorter ones. For example, the following sentences could be divided: p. 10, lines 253-257; p. 11, lines 264-273;lines 274-275.

17.     Please check the grammar and word spelling in the last paragraph of Discussion (added sentences) and the paragraph #4.6 “Bile acids profiling”. For example, the authors added this sentence: “We plan in the next future to deep investigate the contribute of…” looks grammatically incorrect. The more correct version: We plan to investigate the contribution of …. more deeply in the future”.

Author Response

  1. In the Abstract and Introduction, dextran sulphate sodium (DSS) has been replaced with sodium dextran sulfate (lines 26-27, line 81). However, this change was unnecessary since the authors use DSS and not SDS throughout the text. The only mistake that needed to be replaced was “sulphate”. The correct word is “sulfate”.

Answer: the change  performed in the previous version has been eliminated and the word “sulphate” has been replaced with  “sulfate”

  1. There should be more information on why this study is important. In their response to the comments, the authors state that this study “recognizes a relationship between metabolic syndrome, induced by a deregulated diet, and brain BA receptor alterations and neuroinflammation.” However, this rationale is missing in the manuscript. Therefore, it would be great to mention metabolic syndrome and nonalcoholic fatty liver disease (NAFLD) in the manuscript. In the current version, the authors only mentioned NASH, an aggressive form of NAFLD, but not NAFLD in the manuscript.There are no currently approved pharmacological treatments for NASH/NAFLD. This could be also mentioned in the Introduction to further highlight the importance of this research topic.

Answer:  as Reviewer suggested,  we introduced the concept of a  relationship between  alterations  of BA receptors and neuroinflammation  and  metabolic syndrome , of which NAFLD and NASH are hepatic manifestations,  in the introduction and conclusions.  A new reference has been added in the introduction (22).  Steatotic group of mice represented  a NAFLD  model.  So we referred to HFHSD and HFHSD-DSS  groups as NAFLD and NASH respectively. The sentence “there are no currently approved pharmacological treatments for NASH/NAFLD”  has been  added  In the introduction.

  1. If HFHSD-DSS mice represent the model of NASH, then do HFHSD mice represent a model for NAFLD? It needs to be clarified.

Answer:  Steatotic  group of mice, named HFHSD, represent a NAFLD  model, as resulted in our previous work (ref 24 of this paper),  so we referred to HFHSD and HFHSD-DSS  groups as NAFLD and NASH respectively. This concept has been specified in the text and the terms steatotic mice and steatohepatitits  affected mice have been changed  to NAFLD and NASH affected mice.

  1. P. 3, line 124: The authors use “Dentatus gyrus”. It would be more appropriate to use either of these terms: gyrus dentatus (Latin) or dentate gyrus (more common).

Answer:  dentate gyrus  has been used, as suggested by Reviewer.

  1. P. 3, lines 123-124: “…in some districts of Cornu Ammonis (CA)….of hippocampus as compared to controls (Figure 2A and C)”. Could you please clarify what exact hippocampal regions of HFHSD and HFHSD-DSS mouse brains had higher TGR5 expression as compared to the control group? Please also address if Fig. 2C contains data for the sum of TGR5 expressing cells in all three analyzed hippocampal regions or is it an average cell number. Would it not be better to provide graphs for each hippocampal area in Fig. 2C?

Answer: we clarified  what exact hippocampal region of HFHSD and HFHSD-DSS mouse brains had higher TGR5 expression as compared to the control group and graphics for each hippocampal area have been added other than graphic  with the average values for all areas.

  1. P. 4, lines 133-134: “was observed in the CA1, CA3 and DG areas of hippocampus….”. Could you please clarify if FXR levels were significantly higher in all three hippocampal regions? Fig. 3A contains images for different hippocampal areas but the quantification (Fig. 3C) has been performed without distinguishing the areas (Fig. 3C).

Answer: FXR levels were reduced only in HFHSD-DSS group in all the three regions of hippocampus at comparable level so we decided to show a unique graphic  with the average values without distinguish areas.

  1. What exactly cortex area has been analyzed for evaluating TGR5 and FXR expression levels (Figures 2 and 3)? Was it frontal, medial, or caudal? These cortical regions are mentioned in the Materials and Methods (p. 13, line 404).

Answer: All the three areas were analyzed but only in frontal area we found significant difference of FXR and TGR5 expression.  This has been better specified in the text.

  1. It would be better to use HFHSD mice instead of “steatosis affected mice” and HFHSD-DSS mice instead of “steatohepatitis affected mice” for subtitles for consistency and correctness (lines 16-117, 145-146, 181 – subtitles 2.1-2.3).

Answer: This sentences “steatosis affected mice “ and “steatohepatitis affected mice”  have been replaced by HFHSD and HFHSD-DSS mice,  following Reviewer suggestion.

  1. Subtitle 2.5 (p. 9, line 226) needs to be changed. It is well known that Iba1 is a pan-microglial marker whose expression increases with microglial activation. Did authors mean “Alterations in neuroinflammation marker Iba-1 expression in the brains of HFHSD and HFHSD-DSS mice”?

Answer: Subtitle 2.5 has been changed according to Reviewer suggestion

  1. It seems like the full name for UDCA (p.6, line 165) is omitted.

Answer: full name of UDCA has been added

  1. P.7, lines 201-202: It should be endothelial (not epithelial) permeability if the context is BBB. Besides, it would be better to include a reference confirming that PV-1 is a reliable marker of vascular brain injury.

Answer: the word “epithelial” has been changed with “endothelial”. A  reference about PV-1 as a marker of vascular brain injury  has been added (n.25).

  1. It would be good to include reference demonstrating that serum S100b levels can serve as a marker of altered BBB permeability and glial cell activation (p. 9, lines 223-224).

Answer: References on S100beta were already present in the discussion, but one of them has been shifted in result section at the first mention of this protein (n 26).

  1. P. 10, lines 251-252: It would be better to delete “a condition of” since neuroinflammation is a condition itself (characterized by the elaboration of proinflammatory mediators). Besides, “evaluated by the activation of Iba-1” should be replaced with “evaluated by measuring Iba-1 expression”. Iba-1 is increased in activated microglia (it is not activated).

Answer: the sentences have been corrected, according to Reviewer suggestion

  1. It would be more precise to use “To assess neuroinflammation” instead of “To assess this hypothesis” (p.12, line 336).

Answer: the sentence has been corrected, according to Reviewer suggestion

  1. P.12, line 350: It is more correct to use “an increase in the expression of proinflammatory marker Iba-1”. Microglia can be activated but not the marker.

Answer: the sentence has been corrected, according to Reviewer suggestion.

  1. Some sentences are too long and might need to be divided into shorter ones. For example, the following sentences could be divided: p. 10, lines 253-257; p. 11, lines 264-273;lines 274-275.

Answer: all manuscript has been checked  and  long sentences have been divided.

  1. Please check the grammar and word spelling in the last paragraph of Discussion (added sentences) and the paragraph #4.6 “Bile acids profiling”. For example, the authors added this sentence: “We plan in the next future to deep investigate the contribute of…” looks grammatically incorrect. The more correct version: We plan to investigate the contribution of …. more deeply in the future”.

Answer: Last sentence of discussion and paragraph 4.6 have been corrected for spelling and grammar.

All manuscript has been checked for grammar and spelling.